# A Sequential Cross-Sectional Analysis Producing Robust Weekly COVID-19 Rates for South East Asian Countries

**DOI:** 10.3390/v15071572

**Published:** 2023-07-18

**Authors:** Amani Almohaimeed, Jochen Einbeck

**Affiliations:** 1Department of Statistics and Operation Research, College of Science, Qassim University, Buraydah 51482, Saudi Arabia; 2Department of Mathematical Sciences, Durham University, Durham DH1 5JW, UK; jochen.einbeck@durham.ac.uk

**Keywords:** COVID-19, case fatality rates, statistical model, empirical Bayes prediction, shrinkage

## Abstract

The COVID-19 pandemic has expanded fast over the world, affecting millions of people and generating serious health, social, and economic consequences. All South East Asian countries have experienced the pandemic, with various degrees of intensity and response. As the pandemic progresses, it is important to track and analyse disease trends and patterns to guide public health policy and treatments. In this paper, we carry out a sequential cross-sectional study to produce reliable weekly COVID-19 death (out of cases) rates for South East Asian countries for the calendar years 2020, 2021, and 2022. The main objectives of this study are to characterise the trends and patterns of COVID-19 death rates in South East Asian countries through time, as well as compare COVID-19 rates among countries and regions in South East Asia. Our raw data are (daily) case and death counts acquired from “Our World in Data”, which, however, for some countries and time periods, suffer from sparsity (zero or small counts), and therefore require a modelling approach where information is adaptively borrowed from the overall dataset where required. Therefore, a sequential cross-sectional design will be utilised, that will involve examining the data week by week, across all countries. Methodologically, this is achieved through a two-stage random effect shrinkage approach, with estimation facilitated by nonparametric maximum likelihood.

## 1. Introduction

Infectious disease epidemiology has a long history of parameter estimation in mathematical models, often with the aim to interpret and understand the role and impact of parameters in those models (for instance, [1,2]). The study of infectious diseases is a highly interdisciplinary subject combining research from the biological characteristics of the pathogen as well as statistical descriptions of the data behaviour and a mathematical framework to allow for modelling and simulation [3]. Such a mathematical framework can be provided by mixture models. Mixture models have been previously used for different aspects of modelling infectious disease data, including [4], who were interested in modelling prevalence rates and estimating the forces of infection. In a recent preprint, Ref. [5] considered Poisson mixtures to estimate the real-time case fatality rate of COVID-19. The research suggested in this application exploits mixture models through their relationship to random effects, and enables the estimation of robust death rates through empirical Bayes prediction [6].

The idea to use posterior Bayes estimates of random effects in order to stabilise the estimates of mortality rates goes back at least to [7], but has, according to that paper, earlier been presented in some doctoral theses and technical reports, and in related form in the earlier literature on contingency tables. This strand of research draws its motivation from small area estimation problems, where rare events (such as deaths due to certain cancer types) happen in small subpopulations and thus the resulting death rates are based on small counts, and are notoriously unreliable. The principle of [7]’s and similar methods is to robustify the raw, crude rates by taking the death rate from the overall dataset into account. This is put in practice by imposing a random effect for each country or region, that is equipped with some prior distribution (in [7]’s case, a normal prior on the log-odds). After running this through the Bayesian machinery, one obtains ‘shrunk’ rates that can be considered as a compromise of the region-wise and the overall rates, hence reducing variance at a potential increase in bias. The fully Bayesian approach suggested by [7] is rather computationally demanding and based on complex equations. Moreover, the employed Gaussian prior on the log-odds of the rates is not necessarily well suited to account for heterogeneities.

These problems found a solution with the work by [8], who proposed a simple approach for the estimation of random effect models, in which the random effect distribution is approximated (for estimation purposes) by a discrete mixture so that the model, even after the inclusion of covariates, can be fitted by a straightforward EM algorithm. In a satellite paper, Ref. [9] also demonstrated how posterior random effects can be estimated by empirical Bayes shrinkage from the fitted model, closing the circle to [7]’s work. The case for the use of the empirical Bayes methods in this field had been made before by [6].

The work by [10] extended this methodology to two-level scenarios (like male/female subpopulations within countries). This methodology was successfully applied in [11] to obtain robustly estimated rates of mortality due to suicide and intentional self-harm in the Republic of Ireland from 1989 to 1998, using Poisson generalised linear models with region-wise random effects and an appropriate offset representing population sizes. It was also shown in this paper how these robustly estimated rates allow the construction of ‘league tables’ of mortality, and how posterior probabilities of class membership, from the fitted mixture models, can be used to identify clusters or groups of regions of similar behaviour. While one could describe the situation of interest in principle by either Binomial or Poisson models, the latter are somewhat more adequate especially if the number of event counts is small compared to the population sizes. The Poisson GLM with random effect can be seen as a special case of a generalised mixed model [12].

In this work, the object of interest is that of weekly COVID-19 case fatality rates; i.e., weekly death counts by weekly case counts. Both the death or case counts may be small or even zero for certain countries in some weeks. Additionally, we are dealing with the problem of substantial heterogeneity in the death rates between countries. To some part, this heterogeneity is intrinsic, arising from differing host response to the outbreak of the pandemic, or from different testing and vaccination strategies in different countries. Further heterogeneity is added by the way in which the raw data are reported. This “reporting” heterogeneity is mainly, but not only, concerning the denominator, that is the number of infections, which acts as a great unknown. Thus, above all, one could say that we have an ‘uncertain denominator problem’. One can easily show that, if the denominators of rates are equipped with a multiplicative country-wise random effect that captures this heterogeneity, then in a log-linear model, this random effect is just transported into the linear predictor so that it gets absorbed by a usual, additive, random effect, in the sense of the above discussed models. Hence, the models mentioned above are still suitable, with the additional benefit that the mixture approach is particularly well suited to deal with the heterogeneity, and will allow the identification of ‘robust’ clusters of countries with a similar death rate.

Clustering is a powerful method for detecting latent subpopulations or subgroups between observations. The relationship of mixture components and clusters has been discussed, for instance, in [13]. The idea to use mixtures for clustering purposes is much older and has, for instance, already been mentioned in [14], who also studied alternative methods based on densities or distance measures. The computational vehicle to fit mixture models is the expectation maximisation (EM) algorithm. This is a numerical iterative method for approximating the maximum likelihood (ML) estimates of the mixture model parameters. The EM algorithm contains a number of iterations over two steps: an expectation step (E-step) and a maximisation step (M-step). The E-step computes the posterior probabilities that the current unit *i* stems from a certain cluster *k* that will be used in the subsequent M-step to calculate the parameter estimates. The results from each cycle will then be used to update the next cycle and this is continued until the differences in the estimates from the current iteration to the next fall below a small threshold. In the initial E-step, suitable initial parameter values are needed to complete the initial weights. Ref. [15] suggested a grid search for setting the starting values. Different starting values result in different local maxima [16]. This difference in the local maxima depends on whether the EM algorithm has an odd or even number of mass points [8]. Ref. [17] demonstrated that the convergence to a local maximum is assured when the EM algorithm is used to fit mixture models; however, one needs to search over the starting values in order to locate the global maximum. Also, locating the global maximum in fewer EM algorithm iterations depends on the choice of starting values. The effect of the existence of outliers on the performance of the EM algorithm was discussed by [18]. They defined statistical outliers as those observations that differ significantly from the mixture component distributions where the number of components is selected using the starting values.

Ref. [19] stated that, using the EM algorithm to estimate model parameters is preferable due to its generality and simplicity: when the complete data are from an exponential family whose ML estimates are simply calculated, then each M-step of an EM algorithm is also simply calculated. In the mixture density estimation context, Ref. [15] designed the EM algorithm for nonparametric maximum likelihood (NPML) estimation, further developed by [20]. The NPML method provides an approximation of the random effect distribution by a discrete distribution that is based on a finite number of mass points that can be used as intercepts for the different unknown subgroups. A particular advantage of the NPML approach is that the posterior probability corresponds to the weights in the final iteration of the EM algorithm [11]. Another benefit of this methodology is that increasing the number of components locates new mass points without computational effort, which means that there is no restriction for the mass point to be on a grid [8]. Ref. [8] found that the EM algorithm for the NPML estimate of the mixing distribution for both overdispersed and variance component models is “very stable and converged in every case”. Brief details in the context of linear models can be found in [10,21,22].

Selecting the number of clusters in general, or the number of mixture components, *K*, in particular, is a difficult problem. A simple approach, in the context of our model class, is to begin with a standard generalised linear model (K=1, i.e., no random effect) and gradually increase the number of components until the likelihood or a model selection criterion such as AIC or BIC is maximised [10]. Estimating the number of clusters using classical statistical tests is not possible due to the problem of parameter boundary hypothesis; instead, one can use model selection criteria [23]. Ref. [24] showed that unnecessarily large values of *K* may be needed when the NPML estimation is used to maximise the likelihood whereas well-fitting models with a fewer number of classes are usually preferred.

For clarification, it is noted that this work will not investigate the dynamics of the infection over time, but will consider the problem of obtaining more reliable death rates at each fixed point in time. Building on our results, dynamic versions of this methodology can be created in the future, building on compartments and other models as outlined in [3]. With regard to the use of statistical analysis to investigate the dynamics of the infection over time, Ref. [3] remarked that statistics plays an important role in providing meaningful insights into the dynamics of infectious diseases enabling a deep understanding of the patterns and processes.

Section 2 will give a more technical exposition of the problem and the statistical methodology used to address it. Section 3 will present the results of the analysis for the eleven countries of South East Asia, namely Brunei, Cambodia, Timor-Leste (East Timor), Indonesia, Laos, Malaysia, Myanmar (Burma), the Philippines, Singapore, Thailand, and Vietnam. This paper concludes with a discussion in Section 4. Code to reproduce the analysis, in the statistical programming language R, is provided as Appendix A.

## 2. Methododoloy

We applied the methodology recently proposed in [25] and adapted it to the setting of weekly counts. Therefore, in a given (for the moment, fixed) calendar week, $y_{i}$ denotes the observed total number of cases in regions (countries) i=1,…,m, $d_{i}$ denotes the total number of deaths in that week in that region, and $n_{i}$, i=1,…,m denotes the population size of that region. Then, it is clear that crude weekly case-out-of-population rates could be easily computed by $y_{i}/n_{i}$, and crude weekly death-out-of-cases rates (also called case fatality rates in the epidemiological literature) by
(1)$$r_{i}=\frac{d_{i}}{y_{i}}.$$

However, as explained in detail by [25], there are several potential problems with this approach. Firstly, it could be that the observed case rate was 0, making Equation (Equation 1) undefined. Secondly, even if it was non-zero, the count may be small or affected by the measurement error, as alluded to in the introduction. This may lead to highly volatile values of the $r_{i}$. Thirdly, the observed death count $d_{i}$ may be 0 which would result, if defined, in $r_{i}=0$, which may constitute an implausible estimate of the true death rate (referred to as the “latent death rate” in [25]). Finally, there is a time lag between infection and death that certainly causes interpretational problems if the death-out-of-cases rates are considered to reflect any sort of causality. Therefore, one can include a time lag in the computation which has been performed and discussed in detail in [25]. We will not follow this idea here, but only consider un-lagged rates. While this may break the causal link between cases and deaths, it is still of value for monitoring and surveillance purposes as it gives the most current snapshot of the state of the pandemic.

Now, the methodology works by finding “robust” versions of both the numerator and the denominator of (Equation 1). In the case that $y_{i}$ and $d_{i}$ are large, this will leave the crude rate largely unchanged, but if $y_{i}$ and/or $d_{i}$ are small, then the idea is to “borrow” information from other countries in the cross-sectional dataset to arrive at a more reliable rate, for a given fixed week. In statistical terms, this approach aims to reduce variance, at the expense of a moderate increase in bias.

We consider the $y_{i}$ as realisations of random variables $Y_{i}∼\mathrm{Pois}\left(\mu_{i}\right)$, where the expected rates are described by a log-linear model with a random effect (but without any covariates), that is
(2)$$\log\frac{\mu_{i}}{n_{i}}=z_{i}$$
with $z_{i}∼Z$, for some distribution *Z*. The distribution of *Z* will account for the heterogeneity of the population of countries in terms of the expected (reported) COVID-19 counts in a single, given week. Having obtained posterior random effects ${\tilde{z}}_{i}$ from the fitted model (Equation 2), we can compute “shrunk” or robust country-wise case counts via
$${\tilde{y}}_{i}=n_{i}\exp\left({\tilde{z}}_{i}\right),$$
which can be used as a “robust” denominator in (Equation 1).

We build a second model to deal with the numerator. Therefore, the observed deaths $d_{i}$ in country *i*, i=1,…,m are considered as realisations of random variables $D_{i}∼\mathrm{Pois}\left(\lambda_{i}\right)$, where now
(3)$$\log\frac{\lambda_{i}}{{\tilde{y}}_{i}}=c_{i},$$
with random effects $c_{i}$ following a distribution *C* accounting for denominator uncertainty and heterogeneity. After fitting the model, yielding posterior random effects ${\tilde{c}}_{i}$, robust weekly death counts are obtained as
(4)$${\tilde{d}}_{i}={\tilde{y}}_{i}\exp\left({\tilde{c}}_{i}\right)$$
or correspondingly, robust weekly case fatality rates
(5)$${\tilde{r}}_{i}=\exp\left({\tilde{c}}_{i}\right).$$

It remains to specify the random effect distributions *Z* and *C* and the actual estimation methodology. We consider here the nonparametric maximum likelihood approach outlined in several works by Murray Aitkin and co-workers [8,10,17]. Under this approach, these distributions are approximated by discrete mixtures, with *K* and *L* mass points, respectively, the parameters of which are nonparametrically estimated from the data via the EM algorithm. A mixture approach appears particularly suitable here, as it can capture the underlying heterogeneities between countries (in terms of COVID-19 response, cultural or political aspects, geographical proximity, …), and provides an implicit clustering of countries [25] with a view to their death rates. Under this approach, the fitted values ${\tilde{c}}_{i}$, to be plugged into (Equation 5), are then obtained as ‘posterior intercepts’
$${\tilde{c}}_{i}=\sum_{\ell =1}^{L}v_{i\ell }c_{\ell }$$
from the fitted model (Equation 3), where $c_{\ell }$ are the mixture mass points estimated in the EM algorithm and the $v_{i\ell }$ are posterior probabilities, computed via Bayes’ theorem, that country *i* belongs to cluster ℓ=1,…,L. The effect of the shrinkage may be different for different countries, but as a rule of thumb, for most countries, the rates ${\tilde{r}}_{i}$ will be shrunk towards $\exp\left(c_{j(i)}\right)$, where $j\left(i\right)={\arg\max}_{\ell }v_{i\ell }$ is the maximum a posterior estimate of cluster membership for country *i*. Countries with large denominators (here: cases) will have ${\tilde{r}}_{i}\approx r_{i}$ and countries with very small denominators will tend to have shrunk rates which are close to the worldwide (grand) rate of deaths out of cases.

Further details of the estimation methodology under the described two-stage scenario involving models for numerators and denominators have been given in [25] and are therefore not repeated here. It is important to state, however, that the methodology requires the choice of the number of mass points (mixture components) for both *Z* and *C*, which we denote by *K* and *L*, respectively. In [25]’s work, for daily counts, these values were set at K=30 and L=4 based on model selection criteria. For the scenario in the current paper, where we have weekly counts, an equivalent analysis suggested using K=30 and L=6, hence indicating a slightly larger heterogeneity across countries in terms of their death rates. This result is somewhat plausible given other indications in the literature that data aggregation (in this case, from daily to weekly counts), can increase dispersion [26]. It is important to emphasise that the entire methodology described so far is fully cross-sectional. It can, however, be applied sequentially along successive weeks, hence motivating the title of this paper. Such an analysis will be carried out in the following section.

## 3. Results

We extract raw daily counts of new infections and deaths over 234 countries from [27]. We consider the calendar years 2020 to 2022, and firstly aggregate all daily counts to weekly counts (starting Mondays) using this dataset. This then gives us two count data time series (of case counts and deaths) spanning over 159 weeks, for each of the 234 countries. The raw data file also allows one to extract a population size for each country, which is, however, assumed constant across the study period.

We apply the methodology sequentially to a worldwide dataset of counts and deaths for 159 weeks. For the purposes of this paper, we produce and report the robust death rates for only the South East Asia countries, as it would be infeasible to report the results for all 234 countries in a single paper. South East Asia comprises 11 countries: Brunei, Cambodia, Timor-Leste (East Timor), Indonesia, Laos, Malaysia, Myanmar (Burma), the Philippines, Singapore, Thailand, and Vietnam, with a combined population of approximately 680 million people. The countries and corresponding population sizes are displayed in graphical form in Figure 1, with explicit figures given in the first column of Table 1. The largest and most populated country in the region is Indonesia, whereas Brunei has the smallest population.

Firstly, we re-emphasise the fact that crude rates (Equation 1) are generally not computable or not sensible. Table 1 shows, for all eleven countries, in the third column the number of weeks (out of 159) in which the total number of cases was equal to 0, and hence (Equation 1) undefined (NA). The fourth column gives, for all weeks in which (Equation 1) was defined, the number of weeks in which the death count is zero, in which case (Equation 1) is certainly implausible: at any given stage in the pandemic, the risk of dying of COVID-19, once infected, was never 0. We see that both the third and the fourth column of the table contain substantial numbers except for the larger countries, namely Indonesia, Malaysia, and the Philippines.

An interesting example for a situation with many zero counts in both columns is Cambodia. As can be seen in Figure 2, essentially for the whole of 2020 either of $y_{i}$ or $d_{i}$ was equal to zero, hence rendering the raw rates in Cambodia unusable for practically the whole year. A similar picture was again observed in 2022. However, as just stated, even in Cambodia the risk of dying, once infected, was never zero. The methodology described in Section 2, that robustifies the death rates by borrowing information across countries, gives plausible estimates of the latent death rate for the country of Cambodia, even in the year 2020, hovering most of the time around 0.015, as visible from the bottom right panel of Figure 2. Such estimates are then superseded in the year 2021, when case and death numbers rose in Cambodia, allowing reliable estimates of the death rate in their own right. Robust rates then take over again in 2022, albeit now on a much smaller level than in 2020. The bottom left panel of Figure 2 also shows fitted versus raw death rates. It is evident from this figure that many points fall on the diagonal line, indicating that, in this case, no shrinkage has taken place. However, considerable shrinkage has taken place in the situations of zero death rates (corresponding to the pile of points at the left boundary) and for some scattered, very large rates (likely caused by small denominators). It should be added that this plot is not entirely complete, simply for the reason that 26 out of 159 raw death rates do not exist, as can be seen in Table 1.

In Figure 3, Figure 4, Figure 5 and Figure 6, we provide explicit results for Indonesia (the largest country in terms of population size among those considered), Thailand (population being about a quarter of that of Indonesia), Singapore (which is in a somewhat unique position, being a city-state and island-state), and Timor-Leste, which features the second-smallest population size among the 11 countries considered. Analyses for the other six countries are delegated to the Appendix B.

It is noted that the figures do not report fitted *case* rates, as these graphs look virtually indistinguishable from the true case counts (similarly observed in [25]). However, they are not mathematically equal or proportional, since the fitted case rates are always larger than zero. For the fitted *death* rates, we see from the bottom left panels of all figures that the shrinkage can be considerable, but it is notably less strong for very large countries such as Indonesia (Figure 3).

In terms of the development of the pandemic, the picture in Indonesia and Thailand is somewhat similar, with late entries into the pandemic but then two powerful, roughly equally sized spikes in the case numbers in mid-2021 and early 2022. In terms of raw death counts (top right panels), the second spike is for both countries lower than the first spike, and less pronounced for Indonesia than it is for Thailand. The reader may find the grossly differing shapes of the curves in the bottom right panels as compared to the raw death counts surprising. However, this is easily explained by recalling that in the bottom right panels, we see (estimates of) *rates*; i.e., they are subject to a denominator (corresponding to the fitted case counts) which can impact the shapes dramatically. Furthermore, where any of the involved raw counts are small (or zero), the implicit robustification by the model is kicking in here, which may further impact on the shape. Summarising this point, the discrepancies between the two plots on the right hand panels do not indicate poor fits.

Looking now specifically at the fitted death rates in the bottom right panels, we see that, for Indonesia there were two spikes reaching approximately 0.08, roughly corresponding in position to the case spikes (with a slight lag), and reaching in magnitude roughly the death rates observed in the early days of the pandemic. For Thailand, however, the situation is slightly different, with fitted death rates for any of these spikes not exceeding 0.02, staying below the magnitude observed in early 2020. Furthermore, and most notably, the second spike in raw deaths does not correspond at all to any mentionable spike in the fitted death rates; it is rather that the final spike in fitted death rates towards the end of 2022 corresponds to a *third* spike of raw death counts in the upper right panel, which in turn does not seem to relate to any visible spike in the case counts. Finally, it worth mentioning that both countries are quite large so that the overall shrinkage is moderate; the skewed shape of the bottom left panel of Figure 4 is mainly driven by a single, grossly shrunk (but still large) rate, towards the end of 2020.

Proceeding to the smaller countries of Singapore and Timor-Leste, here we see some interesting behaviour with the fitted death rates seemingly decoupled from either case counts or death counts. For Singapore, some apparent spikes in the raw death counts after mid-2021 do not translate into spikes in the fitted death rates; however, a relatively minor but sharp death count spike in early 2021 coinciding with a period of low case counts results in a big spike in the fitted death rates. Upon this occasion, it is worth reiterating that the methodology does not perform any longitudinal smoothing—information is borrowed just across countries, not across time. In Timor-Leste, the generally small raw counts (including many zeros) in the top panels of Figure 6 should be noted, implying that more information needs to be borrowed from other countries. The “background death rate” of just below 0.02 observed here for wide parts of 2020 has a similar interpretation as for Cambodia. This magnitude is not significantly exceeded even at the death count spike of late summer of 2021. The year 2022 sees a mixture of “own” and “background” information, with minor spikes occurring but generally still smallish fitted death rates well below 0.02.

## 4. Discussion

This study used a sequential cross-sectional design, which involved examining a series of cross-sectional data from various time points, to provide valuable insights into the spread of COVID-19 in South-East Asia. The COVID-19 rates for South-East Asian countries have been acquired from the official source; Our World in Data [27]. The robust weekly COVID-19 rates over time were estimated using a random effect model with nonparametric maximum likelihood estimation, and compared across several countries and regions. According to the study, there were a significant number of weeks in which either the case count or the death count was zero, making the crude rates undefined or implausible. This was particularly the case in smaller countries, such as Laos. However, the methodology used in this study was able to robustify the death rates by borrowing information across countries, providing plausible estimates of the latent death rate even in countries with low case and death counts. In Cambodia, for example, the robustified death rate was estimated to be around 0.015 for most of 2020. This is a plausible estimate, as it is below the death rate observed in early 2020, but still above zero.

The fitted death rates differed generally strongly from the shape of the raw death counts; this is firstly due to the random effect shrinkage; but also secondly because the fitted death rates are subject to a denominator, which can have a significant impact on the shape.

The plots of fitted death rates, in the respective bottom right panels of each of Figure 2, Figure 3, Figure 4, Figure 5 and Figure 6 and Figure A1, Figure A2, Figure A3, Figure A4, Figure A5 and Figure A6, allow a broad categorisation of countries according to their fitted death rates over time. For one group of countries, including Brunei, Laos, and Timor-Leste, the fitted death rates fell more or less continuously over the three years of the pandemic, converging to values of almost 0 in 2022 (albeit with some small spikes for Timor-Leste). For a second group of countries, including Cambodia, Singapore, Malaysia, Myanmar, and Vietnam, consistently small fitted death rates were reached in 2022 following major spikes in (typically, late) 2021. A third group of countries, consisting of Indonesia, Thailand, and the Philippines, experienced persistent spikes of the fitted death rates well into 2022.

The major benefit of the methodology applied in this study is that it enables a principled estimation of weekly death (out of cases) rates when the raw rates are mathematically undefined or practically meaningless (due to zero death counts). However, even in periods of the pandemic which are characterised by small case or death counts, it is still vital for every health system to be able to quantify an individual’s risk of dying of COVID-19, once infected. This quantity has implications for resource planning, allocation, and prioritisation, but also for patient consultation or even for simple communication to the “worried well”, hence closing a serious gap in the current computational toolkit underpinning national COVID-19 responses.

The study has some limitations, such as the lack of any temporal smoothing, and the lack of adjustment for slight changes in population size over time. Also, “weekly data” may depend on the chosen cut-off point, and, by construction, may not be able to capture short-term fluctuations in COVID-19 rates.

## Figures and Tables

**Figure 1 viruses-15-01572-f001:**
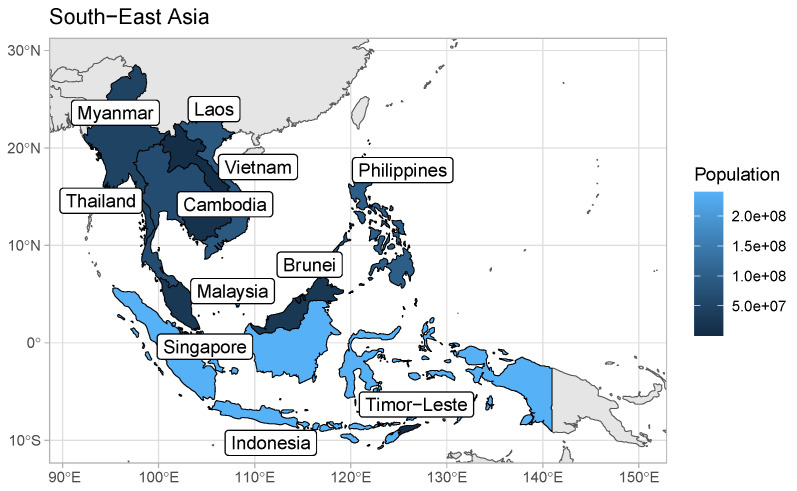
The South East Asian countries. Please note that here and in subsequent figures, the notation ae+b used in legends denotes $a\times 10^{b}$.

**Figure 2 viruses-15-01572-f002:**
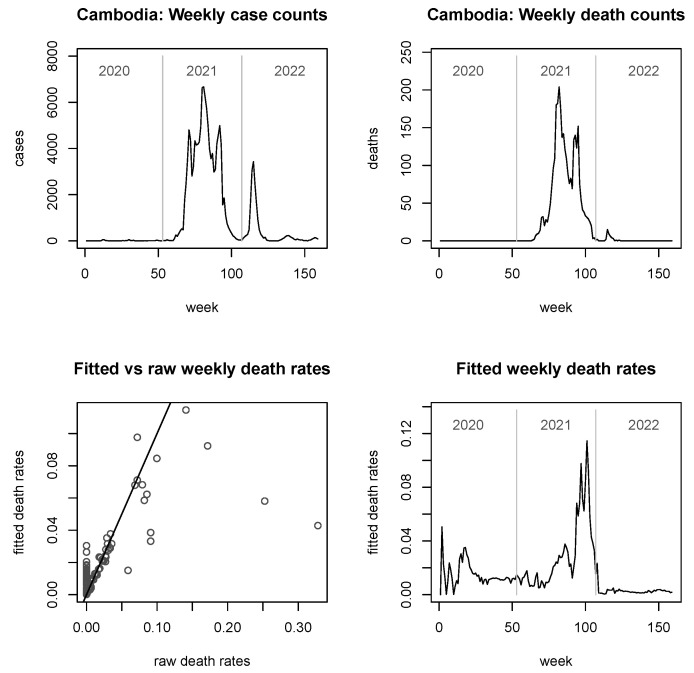
Raw case and death numbers for Cambodia, along with robust, fitted death rates.

**Figure 3 viruses-15-01572-f003:**
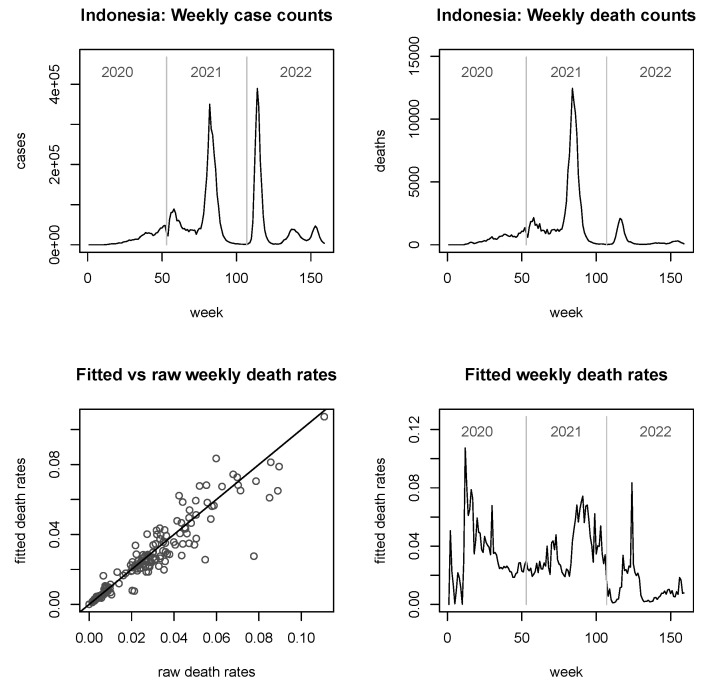
Raw case and death numbers for Indonesia, along with robust, fitted death rates.

**Figure 4 viruses-15-01572-f004:**
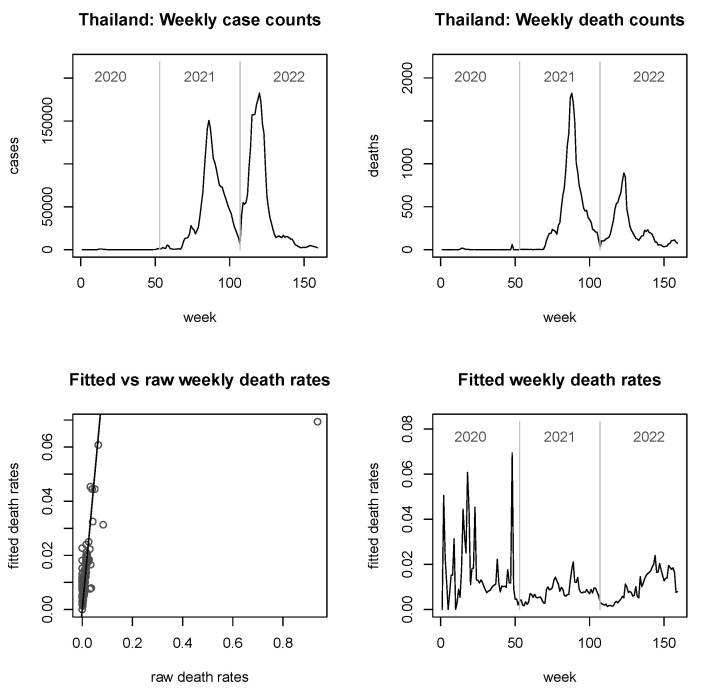
Raw case and death numbers for Thailand, along with robust, fitted death rates.

**Figure 5 viruses-15-01572-f005:**
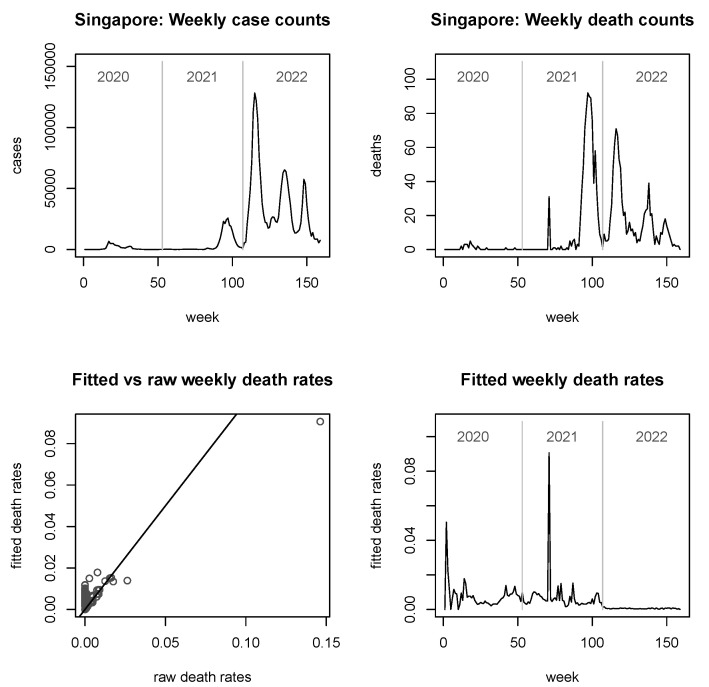
Raw case and death numbers for Singapore, along with robust, fitted death rates.

**Figure 6 viruses-15-01572-f006:**
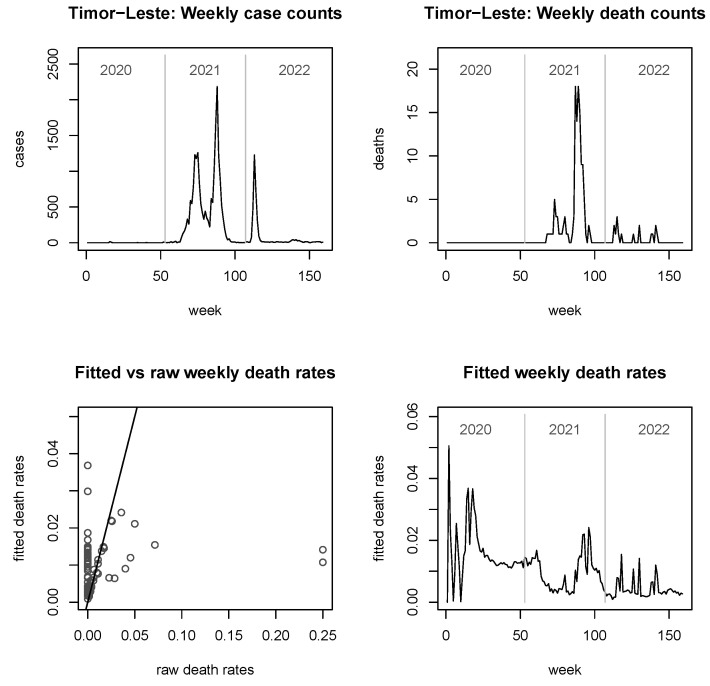
Raw case and death numbers for Timor-Leste, along with robust, fitted death rates.

**Table 1 viruses-15-01572-t001:** For each of the eleven South East Asian countries considered, the population size, the number of times (out of 159 weeks) that the denominator of the weekly crude death rates (Equation 1) is zero, then (otherwise) the numerator is zero, or none of the above occurs.

Country	$n_{i}$	$y_{i}=0$	Otherwise, $d_{i}=0$	Otherwise
Brunei	449,002	32	83	44
Cambodia	16,767,851	26	80	53
Indonesia	275,501,344	9	1	149
Laos	7,529,477	51	70	38
Myanmar	54,179,312	12	48	99
Malaysia	33,938,216	4	16	139
the Philippines	115,559,008	7	3	149
Singapore	5,637,022	3	63	93
Thailand	71,697,024	2	38	119
Timor-Leste	1,341,298	42	79	38
Vietnam	98,186,856	7	60	92

## Data Availability

We make use of publicly available data that can be downloaded from https://covid.ourworldindata.org/data/owid-covid-data.csv (accessed on 1 May 2023). The markdown file in the Appendix A illustrates how our code automatically downloads the data from this source.

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
