# Peer review of "A Sequential Cross-Sectional Analysis Producing Robust Weekly COVID-19 Rates for South East Asian Countries"

_viruses, 2023, doi:10.3390/v15071572_

Round 1

Reviewer 1 Report

This paper presents findings from a sequential cross-sectional study to produce reliable weekly Covid-19 death (out of cases) rates for South East Asian countries for the calendar years 2020, 2021, and 2022. Overall, the analyses appear sound and the findings are interesting. Here are a couple of suggestions for revising the paper.

First, in Figures 2 to 6, it would be good to have clearly stated headings for each of the 4 graphs therein. Yes, most readers can figure this out, but it would be good to state it clearly.

Second, on p. 5, it is stated: "The estimation methodology under the described two-stage scenario involving models for numerators and denominators has been detailed in [6] and is therefore not repeated here." I (and I am sure other readers as well) would feel more comfortable about the results of your analyses if you included a clear statement of the statistical estimation algorithm, including sufficient algebra as necessary for this and also including how you "... "borrow" information from other countries in the cross-sectional data set to arrive at a more reliable rate".

The English grammar is well done.

Author Response

First, in Figures 2 to 6, it would be good to have clearly stated headings for each of the 4 graphs therein. Yes, most readers can figure this out, but it would be good to state it clearly.

The figure headings have been improved. At the request of another reviewer, the bottom left panel of each image has been replaced by a graph showing fitted versus raw death rates. 

Second, on p. 5, it is stated: "The estimation methodology under the described two-stage scenario involving models for numerators and denominators has been detailed in [6] and is therefore not repeated here." I (and I am sure other readers as well) would feel more comfortable about the results of your analyses if you included a clear statement of the statistical estimation algorithm, including sufficient algebra as necessary for this and also including how you "... "borrow" information from other countries in the cross-sectional data set to arrive at a more reliable rate".

We would like to thank the referee for the comment. We have given additional detail, including algebraic expressions, on how “shrunk” rates are computed, and tried to give some intuition in which sense information is “borrowed”. A more detailed description of the inferential machinery of random effect models would be add odds with the style (and audience) of this paper. However, for the interested reader, we do now give full R code to reproduce the results for Cambodia as supplementary material.  The results for the other countries can then be easily reproduced. 

Reviewer 2 Report

The manuscript describes a method for improving estimates of death rates for a particular region by using data from nearby regions. In this case, the authors used countries in southeast Asia as an example. I have some concerns about how this was done.

1. Is is really reasonable to assume that SARS-CoV-2 was spreading in basically the same way in two countries just because they are neighbours? Particularly early in the pandemic, different countries had very different public health measures, but there are also differences in population density and socioeconomic status that could play a large role in both the spread and the number of deaths from COVID.

2. How did the authors decide on the shape of the distribution to pull "data" from? Is there some underlying assumption here or was the shape determined from fits to data?

3. I would like to see graphs of the unfitted death rate alongside the "fitted" death rate. This would allow for a better assessment of what the model is actually doing.

4. I don't think the authors should be using the term "fitting" for their process. Model fitting involves maximizing some kind of likelihood function which is not what the authors are actually doing here.

5. The authors often use 'which' when they should be using 'that'. Please check on the correct usage of these two words.

The English is fine.

Author Response

  1. Is really reasonable to assume that SARS-CoV-2 was spreading in basically the same way in two countries just because they are neighbours? Particularly early in the pandemic, different countries had very different public health measures, but there are also differences in population density and socioeconomic status that could play a large role in both the spread and the number of deaths from COVID.

The methodology does not make any use of two countries being “neighbors”.   Indeed, as the referee says,  the circumstances in each country had been very different, whether neighboring or not.  These different circumstances  lead to heterogeneity, which is captured by the mixture-based random effect applied to both death and case counts.  The “borrowing” of strength then happens entirely through random effect shrinkage. De facto the rates for all countries will be shrunk towards the mass point (cluster) to which they belong, this shrinkage will be smaller for larger countries, and larger for smaller countries.  For countries with very small population size, this shrinkage effect is likely to produce fitted rates which are close to the overall (grand) mean of the worldwide data set. Having said all this, whether or not countries fall into the same cluster (hence having similar fitted rates) will depend on multiple factors, including the ones that the referees mention but also including geographic closeness. This is, however, not explicitly used in the methodology.  We have now given additional detail on how the shrunk rates are obtained, shortly before the end of Section 2.

  1. How did the authors decide on the shape of the distribution to pull "data" from? Is there some underlying assumption here or was the shape determined from fits to data?

We assume that rates on each of the two levels (deaths and cases) are driven by respective latent variables, which we model by random effects. The “shrunk” rates are then obtained as posterior random effects (sometimes called posterior intercepts) based on these models. For the choice of the random effect distribution, there are many options, but given the large heterogeneity between countries,  as well as various but occasionally related circumstances (such as geographical proximity or the use of similar Covid-19 measures), we felt that a mixture approach is a reasonable working assumption. The parameters of that mixture distribution are however fully determined from the data, on both levels. Details in this direction have been added in Section 2.

  1. I would like to see graphs of the unfitted death rate alongside the "fitted" death rate. This would allow for a better assessment of what the model is actually doing.

Thank you for this helpful suggestion. We have now replaced the bottom left panel of each image by the requested plot. We agree that this is much more useful. Some text has been added accordingly to Section 3. Figure headings have also been improved following the suggestion of another referee.

  1. I don't think the authors should be using the term "fitting" for their process. Model fitting involves maximizing some kind of likelihood function which is not what the authors are actually doing here.

The model parameters in a "Nonparametric Maximum Likelihood" (NPML) method are estimated using the Expectation-Maximization (EM) algorithm, which maximizes the likelihood function.

  1. The authors often use 'which' when they should be using 'that'. Please check on the correct usage of these two words.

Thank you. We have corrected these where appropriate.

Reviewer 3 Report

It is a very interesting work and worthy publishing. I just have few remarks:

1. What software  / language did you use fo this work? Is there any scripts or software available to perform these calculations?

2. The results of your work should me more emphasized and exemplified in the Conclusions. It is still unclear for non-professionals what could be brought by your method of calculations and how exactly this changes the landscape of the COVID-19 understanding?

Author Response

  1. What software  / language did you use for this work? Is there any scripts or software available to perform these calculations?

All computations are carried out in the freely available statistical programming language R. We are now providing a supplemental file with R code, and have added this information at the end of the introduction.

  1. The results of your work should be more emphasized and exemplified in the Conclusions. It is still unclear for non-professionals what could be brought by your method of calculations and how exactly this changes the landscape of the COVID-19 understanding?

We agree very much with this comment and have added a new paragraph to the Conclusion (second-last paragraph). The conclusion has also been overworked in other aspects.

Round 2

Reviewer 1 Report

The revisions to this paper have been responsive to the previous review and the manuscript has been improved accordingly. I have no further suggestions for revision. 

Reviewer 2 Report

The authors have addressed my previous comments to my satisfaction.

English is fine.